# Fruit Quality of Several Strawberry Cultivars during the Harvest Season under High Tunnel and Open Field Environments

Hiral Patel [1], Toktam Taghavi [1,*] and Jayesh B. Samtani [2,*]

1   Agricultural Research Station, College of Agriculture, Virginia State University, Petersburg, VA 23806, USA; patelhiral712@gmail.com
2   Hampton Roads Agricultural Research and Extension Center, School of Plant and Environmental Sciences, Virginia Tech, Virginia Beach, VA 23455, USA
*   Correspondence: ttaghavi@vsu.edu (T.T.); jsamtani@vt.edu (J.B.S.)

**Abstract:** Parameters such as titratable acids (TA), total soluble solids (TSS), and their ratio (TSS/TA) are critical in determining strawberry fruit quality and the value of new cultivars. Ten strawberry cultivars were evaluated in two environments (open field and high tunnel) in the city of Virginia Beach. The objective was to evaluate the fruit quality characteristics (total soluble solids, titratable acidity TA, and total anthocyanin content) of newer strawberry cultivars grown in the annual hill plasticulture systems in coastal Virginia climatic conditions. Another objective was to measure the correlation between TA and a new digital meter (pocket acidity meter; PAM). Fruits were harvested weekly and TSS was measured using a refractometer. Acidity was measured using the pocket acidity meter and titratable acidity by a single sample titrimeter. Genetics significantly affected total anthocyanin content, TSS, TA, and acidity. The effect of the environments (high tunnel and open field) was not significant on TSS but significant on total anthocyanin content, TA, and acidity. "Flavorfest" had the highest and "Sweet Ann" the lowest anthocyanin content, TSS, and TA among the cultivars. The acidity (PAM data) showed a different level of correlation than TA, with a higher correlation for the open field than the high tunnel. On average, when outliers were removed, there was a regression of TA = 2.22(PAM) + 0.49 between the two data sets, with $R^2 = 0.47$.

**Keywords:** flavonoids; pigments; protected environment; titratable acidity; acidity; total soluble solids

## 1. Introduction

Strawberries (*Fragaria × ananassa*) are planted for their usually red, sweet, and aromatic fruit. However, due to their short shelf life, worldwide strawberry production is relatively low compared with other fruit crops. Global strawberry production reached 9.22 million metric tons in 2017 [1]. The annual strawberry production in the United States was 1.43 million tons in 2018, accounting for a third of the total world's production, leading the global strawberry market [2,3].

Favorable climate conditions make the states of California and Florida the largest producers of strawberries in the United States [1]. Outside of California and Florida, the South Atlantic is the most productive region (including Virginia) for fresh market strawberries in the United States. There is increasing interest in commercial strawberry production for local markets in Virginia and surrounding states. This region has about 1000 hectares of strawberries [4,5].

Strawberry cultivars are more sensitive to environmental conditions than other fruit crops and their anatomy, morphology, growth habits, and reproductive growth have been studied [6]. However, the relative contribution of genotypic and environmental conditions to fruit quality is not well studied. Additionally, it is unknown to what extent these fruit quality traits remain stable throughout the changing environmental conditions of the cropping season [7].

A high tunnel is a semi-permanent structure with plastic covers that uses passive ventilation for temperature and air humidity adjustment and modifies the environmental conditions to improve fruit quality, extend the season, and protect plants from extreme weather conditions. Strawberries have been cultured in high tunnels or open fields under mild winter climates for out-of-season production. In these cases, specific treatments may secure better performance depending on the cultivar and the environmental conditions [8]. It is known that changes in environmental conditions affect the fruiting capacity and fruit quality, among other characteristics, and can be a source of stress for the crop depending on the ability of the cultivars to cope with it [9]. In this sense, antioxidant compounds of a polyphenolic nature (i.e., anthocyanins) play an essential role in the general mechanisms of the response to different stressors; therefore, changes in environmental conditions could be expected to influence the composition and synthesis of these compounds in fruits. Therefore, different environmental conditions (open field vs. high tunnel) can affect the quality of the fruits in different strawberry genotypes and, consequently, their acceptance by the consumers [7].

Strawberries contain several bioactive phytochemicals, including anthocyanins, flavonols, and phenolic acids. Anthocyanins are a group of flavonoid pigments that are responsible for a wide range of red colors in fruits. Anthocyanins in berries are partly responsible for the high antioxidant activity and have demonstrated a role in protecting plant and human health [10].

Consumers prefer strawberries with a wide range of sensory features. The quality components can be sensory and nutritional [10]. Parameters such as titratable acids (TA), total soluble solids (TSS), and their ratio (TSS/TA) are critical in determining strawberry fruit quality and the value of new cultivars. The TSS to TA ratio is critical in evaluating fruit quality because it determines flavor harmony. Hence, along with fruit color, they are significant factors in determining strawberry fruit quality [10].

High TSS and TA contents represent general selection criteria for the flavor of strawberry fruits in breeding programs. A good and well-balanced flavor for strawberries is based on a high sugar and a comparatively high acid content (i.e., the balance between sweetness and acidity). Their ratio (TSS/TA) is commonly used to evaluate the taste and ripening stage of the fruit. A ratio of (TSS/TA) of 8.5–14 is considered an appropriate balance of sweet–tart flavor notes in strawberries for human palatability [11–13]. In another report, the minimum TSS and maximum TA levels for an acceptable flavor of strawberry were recommended to be 7% and 0.8%, respectively [14,15]. In a third report, the results showed that eating quality was more strongly related to TSS than to TA and a higher TA than recommended (even close to 1%) was still acceptable if combined with high TSS [15].

There are several methods to measure acidity. The pH scale is used to measure the hydrogen ion concentration of a solution. However, pH measurements are not always accurate, especially when dealing with complex or heterogeneous solutions such as fruit extracts.

Titratable acidity measures the hydrogen ions by neutralizing them with sodium hydroxide (base) in a known sample quantity. The amount of the base needed for neutralization reflects the acid content. Compared with pH, titratable acidity is a better predictor of sourness and more closely related to the taste. The process of measuring TA is tedious and labor intensive, even when an automated titrimeter is used, and it needs large amounts of costly reagents, technical expertise, instruments, and a laboratory. A digital meter was recently introduced that quickly measures acidity with minimal preparation and no reagents (pocket acidity meter, PAM F5, Atago, Japan). The PAM measures the acidity level through the electro-conductivity method using electrical current. An electrical current passes through a solution via ions. The conductivity of a solution depends on the concentration of all the ions present and their mobility; however, hydrogens ion are the major contributor to the electrical conductivity of a solution due to their lightweight and faster mobility (on average 10 times more than other ions).

In a previous study, several small fruits' TA and acidity (PAM) levels were measured with an automated titrimeter and a digital meter. A regression was conducted between the two values [16] and the data suggested a strong correlation for blackberries and blueberries, with $R^2$ values of 0.82 and 0.85, respectively. The correlations were not as strong for

raspberries and strawberries and there were some inconsistencies in the data. The reason for the inconsistency was unclear; the $R^2$ values were 0.53 and 0.25 for red raspberry and strawberry, respectively. For the first objective, we expanded the experiment to include more strawberry cultivars and two different environments (high tunnel and open field). The objective was to determine whether the acidity (PAM data) provides a more reliable and precise method to determine the acidity of strawberry extract than titratable acidity.

The second objective of this research was to evaluate fruit quality characteristics (TSS, TA, and total anthocyanin content) of newer strawberry cultivars during harvest season in the annual hill plasticulture system in an open field and under high-tunnel conditions in coastal Virginia climatic conditions (USDA Plant Hardiness Zones 7 and 8). The goal was to determine the effect of the environments and genotypes on strawberry fruit quality during the cropping season.

## 2. Materials and Methods

Strawberries were planted from September 2019 through June 2020 at Hampton Roads Agricultural Research and Extension Center (AREC) in the city of Virginia Beach, Virginia (36°9′ N, 76°2′ W). Strawberries were planted in a Randomized Complete Block Design (RCBD) in four replicates. Soil samples from the top 20 cm were collected before the experiment in both environments and sent to the Virginia Polytechnic Institute and State University Soil Testing Lab. The soil was a tetotum loam with a pH of 5.9 and limestone was broadcast on 5 September 2019, at 1120 kg ha$^{-1}$. The soil pH was adjusted to the desired level of 6.2. The drip irrigation and irrigation systems were set up using a 0.38 mm single drip line with a 30.5 cm emitter spacing (Berry Hill Irrigation, Inc., Buffalo Junction, VA, USA). Fertilizers were applied with pre-plant fertilizer at 69 kg ha$^{-1}$ nitrogen, using Nutrisphere-N (N-P-K ratio of 34-0-0, Southern States Cooperatives Inc., Chesapeake, VA, USA).

A total of ten cultivars ("Camino Real", "Chandler", "Merced", 'Rocco', "Ruby June", "Albion", "Flavorfest", "Keepsake", "San Andreas", and "Sweet Ann") were evaluated in two environments, high tunnel and open field, in a randomized complete block design with four replicates. Short-day cultivars were "Chandler", "Merced", "Camino Real" (all from the University of California, Davis), "Flavorfest" and "Keepsake" (USDA, Beltsville, MD, USA), "Rocco" (North Carolina State University), and "Ruby June" (Lassen Canyon Nursery). Day-neutral cultivars were "Albion", "San Andreas" (UC Davis), and "Sweet Ann" (Lassen Canyon Nursery). Strawberry plugs of all cultivars were ordered from Aaron's Creek Farms Plant Nursery, Buffalo Junction, VA, USA. Fruits were harvested weekly from 2 April until 1 June (Figure 1).

Strawberries were frozen and transferred to the Postharvest Research Lab at Virginia State University (VSU), kept frozen on ice packs during transportation, and placed at −32 °C once they arrived at VSU until further use.

Half of the frozen strawberries were thawed at room temperature, and the juice was used to measure the fruit quality parameters. TheTSS content was measured by a refractometer (Atago, Tokyo, Japan). A few drops of the juice were placed on the refractometer, and the data were presented as °Brix. Acidity was measured by the PAM F5 (pocket acidity meter) from Atago (Tokyo, Japan) and titratable acidity by a single sample titrimeter (EasypH Mettler Toledo, Greifensee, Switzerland). An aliquot of juice or puree was used to prepare a 1:50 solution by adding 1 mL of juice and 49 mL (or g) of distilled deionized water and the solution was mixed. A 0.2 to 0.5 mL aliquot was placed on the refractometer (precalibrated with a 0.04% citric acid solution) using mode 4 (strawberry) setting. The remaining solution was titrated to an endpoint of 8.2 using 0.1 N sodium hydroxide using titrimeters. The TA was calculated based on % citric acid equivalents. Titrimeter readings were plotted against PAM readings, and the linear least squares that fitted the equation with $R^2$ were calculated.

High tunnel          Open field

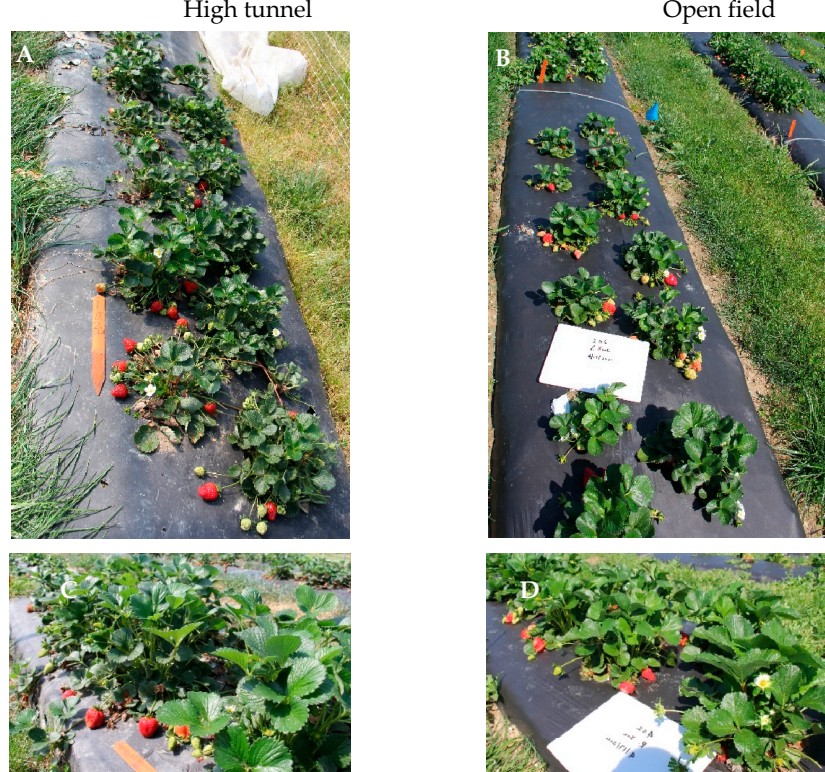

**Figure 1.** Top row, "Camino Real" plants during the harvest season in high-tunnel (**A**) and open-field environments (**B**). Bottom row, "Ruby June" plants during the harvest season in high-tunnel (**C**) and open-field environments (**D**).

Half of the frozen strawberries were sliced and one gram was freeze-dried at $-80\ ^\circ$C (VirTis Freezemobile freeze dryers SP Scientific, Warminster, PA, USA) for anthocyanin extraction. Freeze-dried samples were ground to a fine powder, and anthocyanins were extracted using acidified methanol. Extracts were obtained by adding 20 mL of methanol (acidified with 0.01% HCl) to the strawberry powder. The homogenates were incubated at $4\ ^\circ$C in the dark on a shaker for 24 h. At the end of the incubation period, the homogenates were centrifuged at $4\ ^\circ$C and 7000 rpm for 15 min. The supernatant was then removed and the absorbance of anthocyanin measured using the spectrophotometer at 530 and 657 nm [17]. The anthocyanin concentration was calculated by the following formula and given as A/g fresh fruit tissue (A/gFW), where A = absorbance at 530 and 657 nm, V = volume of extract (mL), and M = fresh mass of the sample (g).

$$Total\ Anthocyanins = \frac{A_{530} - 0.3\ A_{657} \times V}{M} \tag{1}$$

*Experimental Design and Statistical Analysis*

Fruit quality data were analyzed using Proc GLM of Statistical Analysis Software Version 9.4 [18]. For the regression analysis, the Proc REG procedure was used. The data from four replicates were averaged for each experimental unit. The data were averaged for the harvest season to calculate the means for environments or cultivars.

## 3. Results and Discussion

### 3.1. Total Anthocyanins

The total anthocyanin content averaged over the harvest season was lowest in "Sweet Ann" and "Ruby June" (142 and 152 A/gFW, respectively) and highest in "Flavorfest", followed by "Chandler" (289 and 254 A/gFW, respectively) (Table 1). The total anthocyanin

was also affected by the environment and was 23% higher in strawberries grown in the high tunnel than in the open field (Table 1).

**Table 1.** Total anthocyanin content over the growing season of different strawberry cultivars grown in the open field and high tunnel.

| Cultivar | Total Anthocyanin Content (A/gFW) | |
|---|---|---|
| Flavorfest | 289.4 | a * |
| Keepsake | 184.0 | d |
| Rocco | 222.4 | bc |
| Albion | 196.5 | cd |
| Ruby June | 151.8 | e |
| Merced | 187.2 | cd |
| Chandler | 254.1 | b |
| Camino Real | 219.9 | c |
| Sweet Ann | 141.7 | e |
| San Andreas | 212.2 | cd |
| LSD | 32.0 | |
| Environment | | |
| High tunnel | 246.5 | a |
| Open field | 197.7 | b |
| LSD | 16.6 | |

Notes: * Means with the same letters within a column are not significantly different at $p = 0.05$.

When the interaction of cultivars and the environments were analyzed, "Flavorfest" grown in the high tunnel had the highest total anthocyanin content (300 A/gFW), which was 127% higher than "Sweet Ann" (132 A/gFW) in the open field (Table 2). A higher anthocyanin content has been reported for strawberries and raspberries grown in the high tunnel compared with those grown in the open field [19,20].

**Table 2.** The interaction of environment and cultivar on the total anthocyanin content (A/gFW) of strawberries.

| Environment | Cultivar | Total Anthocyanin Content (A/gFW) | Standard Error |
|---|---|---|---|
| | Flavorfest | 300.0 a * | 10.16 |
| | Keepsake | 212.7 bcd | 19.80 |
| | Rocco | 188.9 bcd | 23.66 |
| | Ruby June | 165.3 cd | 25.56 |
| High Tunnel | Merced | 172.9 cd | 25.56 |
| | Chandler | 193.7 bcd | 31.30 |
| | Sweet Ann | 211.7 bcd | 31.30 |
| | San Andreas | 228.7 abc | 62.60 |
| | Avg | 209.3 | |
| | Flavorfest | 208.8 bcd | 28.00 |
| | Keepsake | 161.9 cd | 17.36 |
| | Rocco | 229.7 abc | 11.07 |
| | Albion | 196.5 bcd | 11.63 |
| | Ruby June | 148.6 cd | 12.52 |
| Open Field | Merced | 190.1 bcd | 11.63 |
| | Chandler | 263.4 ab | 12.28 |
| | Camino Real | 219.9 bc | 11.63 |
| | Sweet Ann | 132.0 d | 11.63 |
| | San Andreas | 211.6 bcd | 11.63 |
| | Avg | 196.2 | |

Notes: * Means with the same letters within a column are not significantly different at $p = 0.05$. Fruits for cultivars "Albion" and "Camino Real" in the high tunnel were not available for analysis.

### 3.2. Total Soluble Solids

Total soluble solids (TSS) content was highest in "Flavorfest" (9.65 °Brix) and "Keepsake" and lowest in "San Andreas" (7.16 °Brix) when averaged over the two environments (Table 3). The total soluble solids content for "Rocco", "Albion", and "Ruby June" were not significantly different than "Flavorfest" and "Rocco". TSS content was not significantly different between the strawberries grown in the high tunnel and open field; both were above 8 (°Brix) when averaged over the cultivars.

**Table 3.** The total soluble solids (TSS) content, acidity, and titratable acidity (TA) of strawberry cultivars averaged over the harvest season and in the high-tunnel or open-field environments.

| Cultivar | TSS (°Brix) | Acidity (%, PAM) | TA (%) | TSS/TA |
|---|---|---|---|---|
| Flavorfest | 9.65 a * | 0.67 a | 1.94 a | 4.97 |
| Keepsake | 9.20 a | 0.63 ab | 1.79 ab | 5.14 |
| Rocco | 8.83 ab | 0.63 ab | 1.93 a | 4.58 |
| Albion | 8.82 ab | 0.58 cd | 1.77 abc | 4.98 |
| Ruby June | 8.79 ab | 0.60 bc | 1.77 abc | 4.97 |
| Merced | 8.22 bc | 0.51 e | 1.52 d | 5.41 |
| Chandler | 7.93 bcd | 0.57 cd | 1.79 ab | 4.43 |
| Camino Real | 7.59 cd | 0.49 e | 1.60 cd | 4.74 |
| Sweet Ann | 7.56 cd | 0.51 e | 1.54 d | 4.91 |
| San Andreas | 7.16 d | 0.55 d | 1.64 bcd | 4.37 |
| LSD | 0.87 | 0.04 | 0.16 | |
| Environment | | | | |
| High Tunnel | 8.08 a | 0.62 a | 1.76 a | 4.59 |
| Open Field | 8.30 a | 0.50 b | 1.64 b | 5.06 |
| LSD | 0.32 | 0.14 | 0.06 | |

* Means with the same letters within a column are not significantly different at $p = 0.05$.

The TSS values in this study are slightly lower than previous reports but within the acceptable range of 7–12% reported by the Oregon Strawberry Commission [11] and Keutgen and Pawelzik [12]. For example, "Korona" and "Elsanta" had TSS contents of 9.5 and 8.4%, respectively [14].

### 3.3. Acidity and Titratable Acidity

The acidity (PAM) values were lower than the TA values measured using the titrimeter. The PAM values were highest in "Flavorfest", "Keepsake", and "Rocco" and lowest in "Sweet Ann", "Merced'", and "Camino Real". The TA showed the same pattern: highest in "Flavorfest" and "Rocco" and lowest in "Sweet Ann" and "Merced" (Table 3). The differences among cultivars were subtle for TA values. Both values were higher in strawberries grown in the high tunnel than in the open field.

Genotype had substantial effects on the TSS and TA. Zhang et al. [21] stated that both TSS and TA are strongly influenced by genotype and that the TSS content was particularly high in "Sabrina", "Rubygem", "Sabrosa", and "Camarosa". Herrington et al. [22] and Saraçoğlu [23] found that altitudes had a weak effect on TSS. Gündüz and Özbay [13] also reported that genotype substantially affected the TSS and TA but that location/altitude had a weak effect on TSS. Similar results were reported by Andreotti et al. [24] for different altitudes in South Tyrol (Italy). Herrington et al. [22] found that the genotype effect was more significant than the growing location for TA. Gündüz et al. [13] reported that the TA content in strawberries varied more intensely due to fruit maturity, genotype, and nutrition than ecological factors. Our data also confirm that the TSS varied more intensely due to the genotype than due to the open-field or high-tunnel environments. Changes in TA and acidity (PAM data) were more profound in different cultivars than in the outdoor or high-tunnel environments.

### 3.4. TSS/TA Ratio

The TSS/TA ratio is generally recommended as a quick measure of consumer acceptance [12]. The ratio varied from 4.4 to 5.4 in different cultivars and environments. The ratios in this experiment were lower than those previously reported, mainly due to the higher TA values [12,15]; however, these ratios did not correspond with any off-flavor taste.

The recommended TA is a maximum of 0.8%, whereas the minimum recommended TSS is 7% for an acceptable flavor in strawberries [14]. However, it was reported that eating quality was more strongly related to TSS than to TA [25]. A higher TA than recommended (even close to 1%) was still acceptable if combined with a high TSS content. Even so, an average sugar/acid ratio of 5.3 for the "Oso Grande", "Toyonoka", and "Mazi" cultivars was adequate to achieve the best quality [26]. Our results showed higher TA levels (close to 1) and a lower TSS/TA ratio than the recommendation [14] but with an acceptable strawberry flavor. Our data also confirmed that the TSS content has a more profound effect on eating quality than the TA.

### 3.5. TSS, Acidity, and TA over the Harvest Season

The TSS in strawberries did not change between the high-tunnel and open-field environments during the harvest season, except on the second harvest date, which was higher under the high tunnel than in the open field (Table 4). "Flavorfest" had a higher TSS content than the other cultivars, followed by "Keepsake". "San Andreas", "Sweet Ann", "Camino Real", and "Chandler" are among those that have the lowest TSS contents over the harvest season.

**Table 4.** Total soluble solids (TSS) of strawberry cultivars during the first five harvest dates and in high-tunnel or open-field environments.

| Cultivar | °Brix-Avg | °Brix-Harvest1 | °Brix-Harvest2 | °Brix-Harvest3 | °Brix-Harvest4 | °Brix-Harvest5 |
|---|---|---|---|---|---|---|
| Flavorfest | 9.98 a * | 9.76 a | 9.10 a | 10.13 a | - [b] | - |
| Keepsake | 9.22 b | 8.93 b | 8.90 a | 9.28 b | 11.23 a | - |
| Rocco | 8.80 b | 8.55 bc | 8.35 b | 8.23 cd | 8.63 b | 8.79 ab |
| Albion | 8.70 bc | 7.30 e | 7.48 c | 8.47 bc | 7.93 cd | 8.53 b |
| Ruby June | 8.79 b | 8.15 cd | 7.49 c | 9.31 b | 8.89 b | 9.28 a |
| Merced | 8.21 cd | 8.12 cd | 7.60 c | 8.09 d | 8.41 bc | 8.68 b |
| Chandler | 7.94 de | 7.84 d | 7.65 c | 7.32 e | 7.23 e | 8.51 b |
| Camino Real | 7.70 de | 7.27 e | 7.20 c | 7.31 e | 7.89 cd | 7.76 c |
| Sweet Ann | 7.59 ef | 7.08 e | 7.27 c | 7.88 de | 7.68 de | 7.47 c |
| San Andreas | 7.14 f | 7.06 e | 7.11 c | 6.61 f | 7.78 de | 7.32 c |
| LSD | 0.52 | 0.44 | 0.50 | 0.61 | 0.59 | 0.52 |
| | | | | | | |
| Environment | | | | | | |
| High Tunnel | 8.42 a | 7.99 a | 7.88 a | 8.16 a | 8.13 a | 8.34 a |
| Open Field | 8.30 a | 7.93 a | 7.56 b | 8.03 a | 8.11 a | 8.24 a |
| LSD | 0.23 ns | 0.20 ns | 0.20 ** | 0.22 ns | 0.26 ns | 0.25 ns |

Notes: * Means with the same letters within a column are not significantly different at $p = 0.05$. [b] No data were collected during those weeks as the season for certain cultivars ended. ns means not significant and ** means significant at $p = 0.01$.

Titratable acidity and acidity (PAM data) were consistently higher in the high tunnel than in the open field strawberries during the harvest season (Tables 5 and 6). The TA value was highest in "Flavorfest" on the first harvest date; however, as we moved through the season, "Keepsake" and "Rocco" exhibited higher TA values. "Merced" had the lowest TA value throughout the harvest season (Table 6). Acidity (PAM) followed the same pattern, with "Flavorfest" having higher acidity and being replaced by "Keepsake" as we moved through the harvest season. The lowest acidities were measured in "Camino Real", "Merced", and "Sweet Ann" throughout the harvest season (Table 5).

**Table 5.** Acidity (%) of strawberry cultivars during the first five harvests and in high-tunnel or open-field environments (PAM data).

| Cultivar | Acidity-Avg | Acidity-Harvest1 | Acidity-Harvest2 | Acidity-Harvest3 | Acidity-Harvest4 | Acidity-Harvest5 |
|---|---|---|---|---|---|---|
| Flavorfest | 0.71 a * | 0.72 a | 0.55 c | 0.65 a | - [b] | - |
| Keepsake | 0.64 b | 0.69 a | 0.62 ab | 0.56 bc | 0.71 a | - |
| Rocco | 0.63 b | 0.63 b | 0.67 a | 0.58 b | 0.61 b | 0.58 b |
| Albion | 0.59 c | 0.56 cde | 0.55 c | 0.60 ab | 0.56 bc | 0.59 ab |
| Ruby June | 0.59 c | 0.60 bcd | 0.58 bc | 0.61 ab | 0.55 c | 0.63 a |
| Merced | 0.51 fg | 0.53 ef | 0.47 d | 0.50 cd | 0.45 de | 0.53 c |
| Chandler | 0.56 cd | 0.60 bc | 0.59 bc | 0.58 b | 0.53 cd | 0.50 c |
| Camino Real | 0.48 g | 0.49 f | 0.44 d | 0.48 d | 0.47 e | 0.50 c |
| Sweet Ann | 0.53 ef | 0.54 def | 0.54 c | 0.51 cd | 0.44 e | 0.51 c |
| San Andreas | 0.55 de | 0.57 cde | 0.56 bc | 0.55 bcd | 0.56 bc | 0.51 c |
| LSD | 0.03 | 0.05 | 0.06 | 0.06 | 0.05 | 0.04 |
| | | | | | | |
| Environment | | | | | | |
| High Tunnel | 0.65 a | 0.65 a | 0.64 a | 0.63 a | 0.62 a | 0.63 a |
| Open Field | 0.50 b | 0.52 b | 0.48 b | 0.48 b | 0.45 b | 0.47 b |
| LSD | 0.01 | 0.02 | 0.02 | 0.02 | 0.02 | 0.02 |

Notes: * Means with the same letters within a column are not significantly different at *p* = 0.05. [b] No data were collected during those weeks as the season for certain cultivars ended.

**Table 6.** The titratable acidity (TA, %) of strawberry cultivars during the first five harvests and in high-tunnel or open-field environments.

| Cultivar | TA-Avg | TA-Harvest1 | TA-Harvest2 | TA-Harvest3 | TA-Harvest4 | TA-Harvest5 |
|---|---|---|---|---|---|---|
| Flavorfest | 2.04 a * | 2.00 a | 1.85 ab | 1.85 ab | - [b] | - |
| Keepsake | 1.80 c | 1.91 ab | 1.68 bc | 1.80 ab | 1.68 bc | - |
| Rocco | 1.93 b | 1.86 abc | 1.97 a | 1.74 abc | 2.10 a | 1.81 a |
| Albion | 1.80 c | 1.85 abc | 1.77 ab | 1.85 ab | 1.52 bcd | 1.82 a |
| Ruby June | 1.77 c | 1.67 bcd | 1.85 ab | 2.04 a | 1.59 bcd | 1.71 ab |
| Merced | 1.52 e | 1.62 cd | 1.47 c | 1.36 c | 1.59 bcd | 1.53 bc |
| Chandler | 1.78 c | 1.95 a | 1.77 ab | 1.70 abc | 1.72 b | 1.60 abc |
| Camino Real | 1.60 de | 1.69 bcd | 1.68 bc | 1.73 abc | 1.37 cd | 1.36 c |
| Sweet Ann | 1.57 de | 1.57 d | 1.70 abc | 1.53 bc | 1.31 d | 1.63 ab |
| San Andreas | 1.65 d | 1.75 abcd | 1.82 ab | 1.52 bc | 1.76 b | 1.60 abc |
| LSD | 0.10 | 0.22 | 0.24 | 0.34 | 0.31 | 0.24 |
| | | | | | | |
| Environment | | | | | | |
| High Tunnel | 1.83 a | 1.84 | 1.86 a | 1.75 a | 1.70 a | 1.74 a |
| Open Field | 1.64 b | 1.72 | 1.64 b | 1.64 a | 1.56 b | 1.54 b |
| LSD | 0.04 | 0.10 | 0.10 | 0.12 ns | 0.13 | 0.12 |

Notes: * Means with the same letters within a column are not significantly different at *p* = 0.05. [b] No data were collected during those weeks as the season for certain cultivars ended.

### 3.6. TA and Acidity (PAM Data) Regression

Titratable acidity estimates the sourness and sweetness of a fruit, but its measurement is labor intensive, expensive, and tedious, even when an automated titrimeter is used. Therefore, a pocket-sized digital meter (PAM) that quickly measures TA with minimal preparation was trialed for different cultivars of strawberries in two environments: high tunnel and open field. The regression data (TA and acidity) averaged over different cultivars and the two environments (high tunnel and open field) were significant at $p \leq 0.01$, with a linear least squares fit equation of 1.66X + 0.77 and $R^2$ = 0.26. The data show a weak correlation between the TA and acidity (PAM) data (Figure 2A); however, when the outliers were removed (Figure 2B), the $R^2$ increased to 0.47, with the equation of 2.22X + 0.49, where X is the acidity.

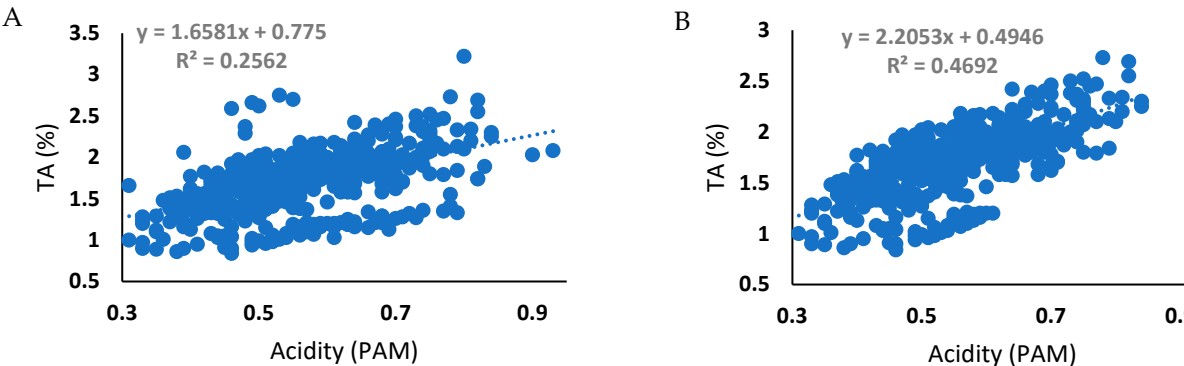

**Figure 2.** The regression between acidity measured using the PAM and the titratable acidity (TA) measured using titration with NaOH 0.1 N. (**A**) All data were included in the graph; (**B**) outliers were removed. The $R^2$ increased to 0.4692 when the outliers were removed.

The regression data for each cultivar in each environment showed that, in high tunnels, cultivars were different, either not showing any correlation between acidity (PAM) and TA, such as for "Chandler" ($R^2 = 0.0004$), or showing a medium level of correlation, such as for "Rocco" ($R^2 = 0.32$). However, in the open field, all cultivars except "Rocco" showed more correlation than the high tunnel and $R^2$, ranged between 0.20 and 0.38 (Figure 3). For example, "Chandler" showed a much higher correlation ($R^2 = 0.31$) between acidity (PAM) and TA content in the open field compared with the high tunnel. The cultivar "Ruby June" had the highest correlation between TA and acidity (PAM) in the open field, with a linear least squares fit equation of 2.35X + 0.49 and an $R^2$ of 0.38.

Our results indicated that the pocket meter can be a very rapid means of estimating titratable acidity in strawberries. The results confirmed the previous report by Perkins-Veazie et al. [17] that the correlations between TA and acidity (PAM data) were not strong for strawberries and there were some inconsistencies. Our results show that acidity data were more consistent for strawberries grown in the open field than for those grown in the high-tunnel environment. There are no solid results on cultivars, with some showing a low correlation in the high tunnel and a higher correlation in the open field. The influence of harvest date or organic acid profile of the strawberry germplasm may be essential for further studies.

Ideally, the PAM data should match the titrimeter values. This was not always the case for these data and we could not predict what caused the outliers within each cultivar of strawberries or the environment. The PAM data had only two decimal places, whereas the automated titrimeters yielded four. If very precise values for TA are needed, such as when developing value-added products, then an automated titrimeter would be the better choice. An added attraction is that two or three of the PAM meters could be used by several people at minimal cost and further accelerate the collection of titratable acidity data. Further, using a PAM does not require special chemicals or a specialized laboratory.

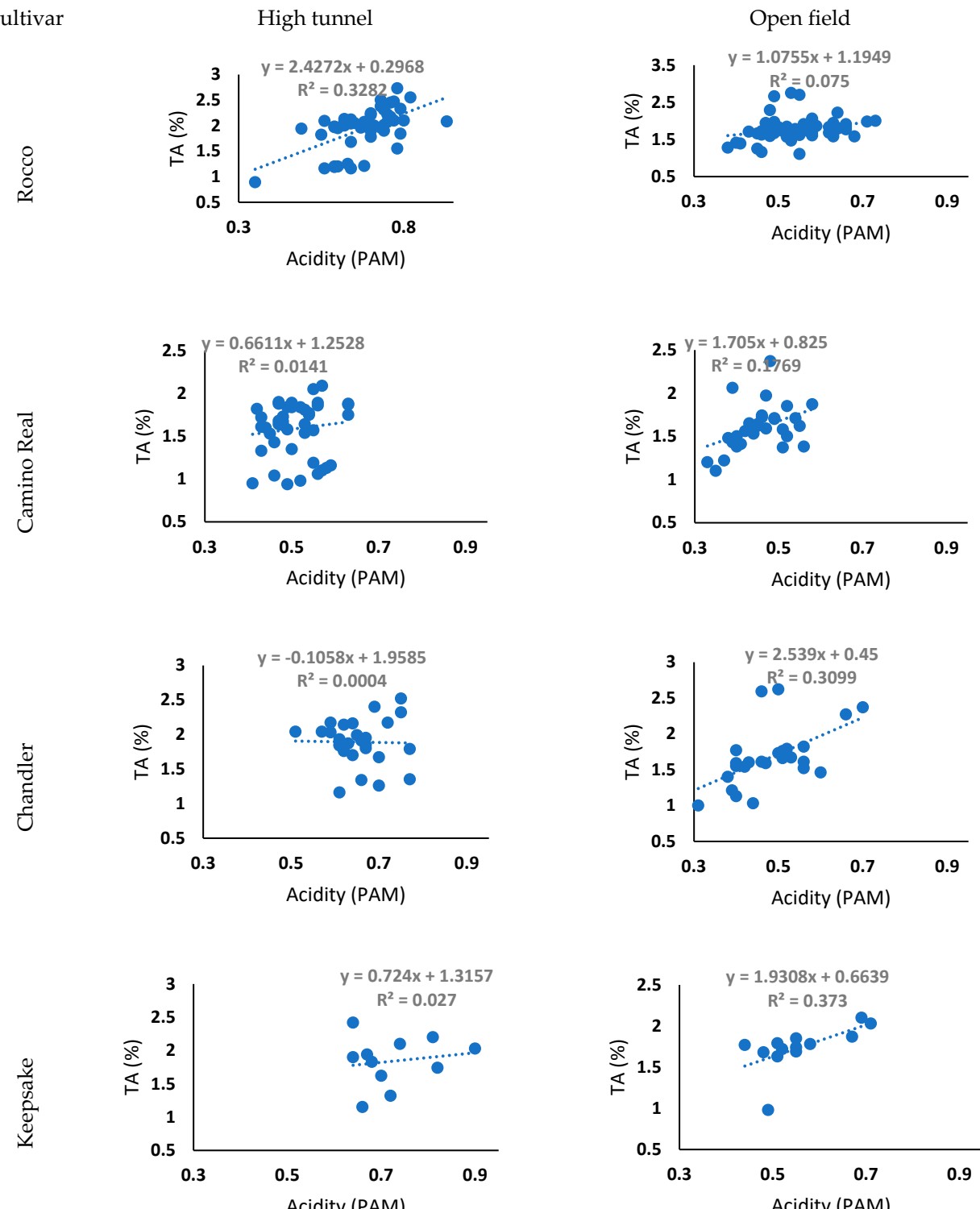

**Figure 3.** *Cont.*

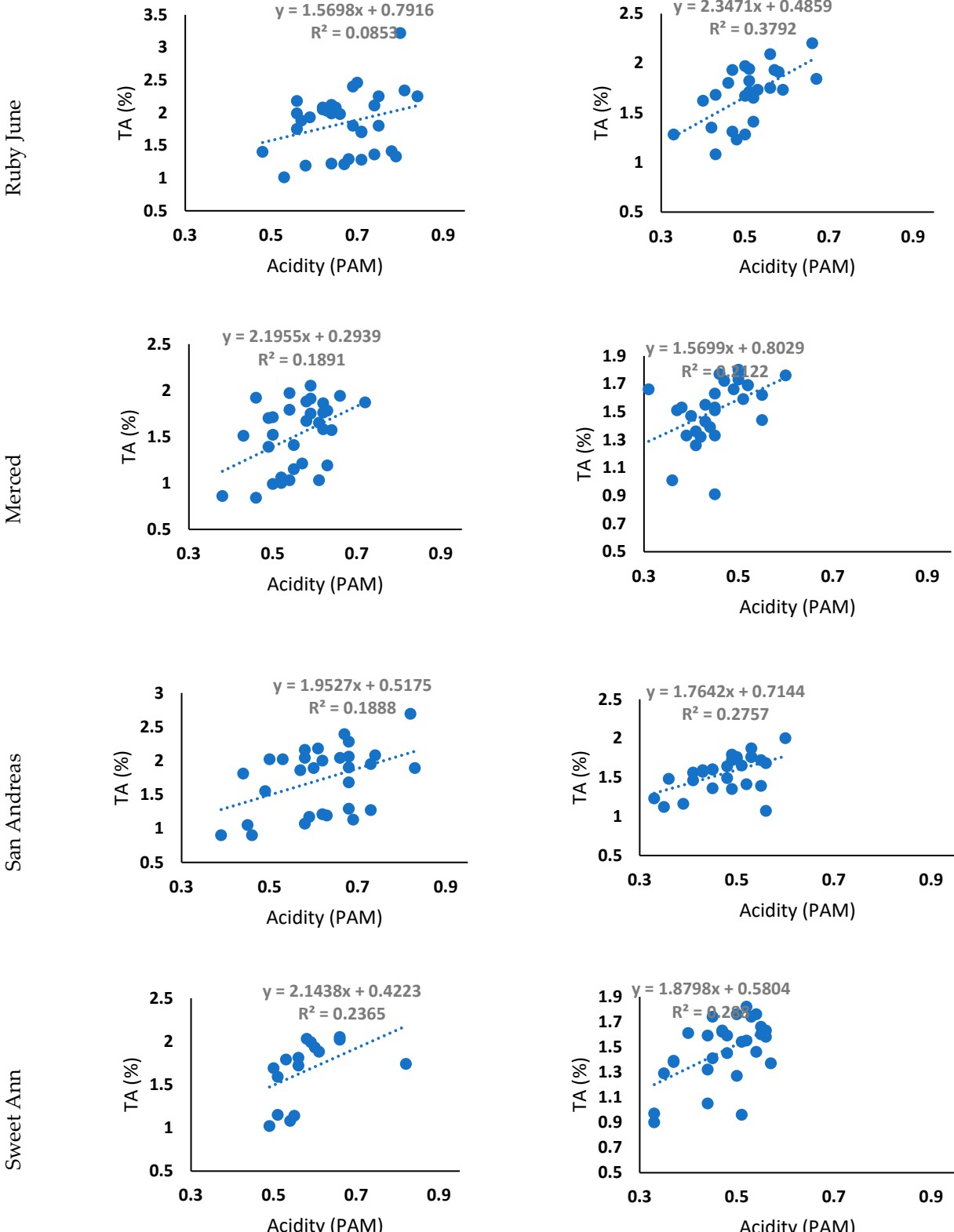

**Figure 3.** The regression between acidity (PAM data) and the titratable acidity (TA) measured by titration with NaOH 0.1 N for cultivar and environment interactions. The cultivars "Flavorfest" and "Albion" did not have enough representations in the samples.

## 4. Conclusions

Genetics substantially affects strawberry fruit quality (total soluble solids, TA, and acidity) more than the two environments. Strawberry cultivars had higher anthocyanin content and acidity in the high tunnel than in the open field. The environment (high tunnel and open field) did not have a significant effect on TSS. Anthocyanin and TSS contents were higher in "Flavorfest" than in other cultivars. "Sweet Ann" and "San Andreas" had the lowest TSS and TA. The fruit quality parameters did not significantly change during the harvest season.

Although the correlation between the TA and acidity (PAM data) was not strong, the TA can be estimated using PAM efficiently and with little effort. The estimate will fulfill the industry requirements for identifying the harvest date (especially in the open field) considering that the acidity level is not a strong indicator of fruit quality. The reason for the weak correlation is not understood and warrants further investigation.

**Author Contributions:** Conceptualization, methodology, software, validation, resources, data curation, visualization, supervision; and funding acquisition, T.T. and J.B.S.; formal analysis, T.T.; investigation, H.P., T.T. and J.B.S.; writing-first draft T.T.; supervision, T.T. All authors have read and agreed to the published version of the manuscript.

**Funding:** This research was funded by the USDA Specialty Crop Block Grant, USDA-NIFA-Capacity Building Grant (grant number 2019-38821-29038) and the USDA-NIFA-Evans Allen (grant numbers 1015372). Virginia State University provided staff salary and support.

**Data Availability Statement:** All available data are presented in the manuscript.

**Acknowledgments:** The authors thank Danyang Liu, Aman Rana, Sophia Gonzales, Robert Holtz, and Greyson Dockiewicz for their assistance in field plot maintenance. This article is a contribution of the Virginia State University (VSU) Agricultural Research Station (Journal Article Series Number 394).

**Conflicts of Interest:** The authors declare no conflict of interest.

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
