# Peer review of "Fruit Quality of Several Strawberry Cultivars during the Harvest Season under High Tunnel and Open Field Environments"

_horticulturae, doi:10.3390/horticulturae9101084_

Round 1

Reviewer 1 Report

1. The high edible value of strawberries and their cultivation in various parts of the United States were emphasized and the quality of strawberry was also described, but the purpose and significance of the research were not clearly stated. Furthermore, the research with the title of total anthocyanins and fruit quality of strawberry cultivars are affected by high tunnel and open field conditions” will study the effect of different conditions on the strawberries, the author should write some conditions on the strawberry or other horticultural products. However, the kind of content does not include in the introduction.

2. In materials and methods, the interval time between the 5 harvesting periods in Table 4 was not clearly explained. Also, what is the high tunnel? What is the open-field? There is no explanation in the manuscript.

3. Why is not labeled with letters in Table 2?

4. What is the difference between acidity and titratable acidity?

5. What is the unit of the acidity?

6. The experimental design is good with ten varieties strawberry and two treatments, however, the measured quality characteristic indicators are too few, there are a total of four indicators. The author should add some other relevant indicators. As for the mentioned Pocket Acid Meter (PAM F5, Atago, Japan) instrument that can quickly measure TA and acid, it is not the innovation of this study. As for the correlation between the study of titratable acidity and acid, it is not significant on strawberries and no convincing results have been obtained.

7. If the manuscript includes some pictures with different cultivars in two cultivation conditions will do better.

Author Response

  1. The high edible value of strawberries and their cultivation in various parts of the United States were emphasized and the quality of strawberry was also described, but the purpose and significance of the research were not clearly stated. Furthermore, the research with the title of “total anthocyanins and fruit quality of strawberry cultivars are affected by high tunnel and open field conditions” will study the effect of different conditions on the strawberries, the author should write some conditions on the strawberry or other horticultural products. However, the kind of content does not include in the introduction.

A: The following paragraphs were added to the introduction to add the suggested content.

“Strawberry varieties are more sensitive to environmental conditions than other fruit crops and their anatomy, morphology, growth habit, and reproductive growth have been studied ((Larson 2018)). However, the relative contribution of genotypic and environmental conditions on fruit quality is not well studied. Additionally, it is unknown to what extent these fruit quality traits remain stable throughout the changing environmental conditions during the cropping season (Cervantes et al, 2020).

Strawberry culture may be practiced in protected conditions or open field under mild winter climates for out of season production. In these cases a better performance may be secured by specific treatments, depending on the cultivar and the environmental conditions (Paroussi et al., 2002).     It is known that changes in environmental conditions affects fruiting and fruit quality, among other characteristics and can be a source of stress for the crop depending on the ability of the varieties to cope with it. In this sense, antioxidant compounds of polyphenolic nature play an important role in the general mechanisms of response to different stressors and, therefore, changes in environmental conditions could be expected to influence the composition and synthesis of these compounds in fruits. Therefore, different environmental conditions (open field vs high tunnel) can affect the quality of the fruits in the different strawberry genotypes, and, consequently, their acceptance by the consumers (Cervantes et al, 2020). “

  1. In materials and methods, the interval time between the 5 harvesting periods in Table 4 was not clearly explained. Also, what is the high tunnel? What is the open-field? There is no explanation in the manuscript.

A: A sentence was added to Materials and Methods to describe the harvest date intervals. This sentence was added to the introduction to define high tunnels.

“High tunnel is a semi-permanent structure with plastic covers which uses passive ventilation for temperature and air humidity adjustment and modifies environmental conditions, to extend the season, and protect plants from extreme weather conditions. Strawberries has been cultured in high tunnels or open field under mild winter climates for out of season production.”

  1. Why is not labeled with letters in Table 2?

A: Letter grouping was added to Table 2.

  1. What is the difference between acidity and titratable acidity?

A: Four paragraphs were added to the introduction to define the acidity and the different methods for measuring them. Below is the full explanation:

“Strawberry acidity comes mainly from citric acid which comprises about 88% of the acid content, along with malic acid and ellagic acid. When strawberry ripens, the acidity decreases. There are several methods to measure acidity.

The pH scale is used to measure the hydrogen ion concentration of a solution. However, pH measurements are not always accurate, especially when dealing with complex or heterogeneous solutions such as fruit extracts.

Titratable acidity measures the total available hydrogen ions in a known quantity of the sample by neutralizing them with sodium hydroxide (base). The amount of the base needed for neutralization reflect the acid content. Titratable acidity (%) provides an estimate of acids in fruits and compared to pH is a better predictor of acids that impact flavor. Traditionally, 

The total acidity is the sum of all the organic acid anions in solution. PAM (ATAGO Brix-Acidity Meters) measures the acidity level through electro-conductivity method using electrical current. There is a relation between the pH and the electrical current flow [the conductivity] of a solution. Electrical current passes through a solution by ions. The conductivity of a solution depends on the concentration of all the ions present and their mobility; however, hydrogen ion is the major contributor to the electrical conductivity of a solution due to its lightweight and faster mobility (on average 10 times more than other ions).

The objective was to determine whether the conductivity measurements provide a more reliable and precise method to determine the acidity of strawberry extract than titratable acidity.”

  1. What is the unit of the acidity?

A:  The unit for acidity is %. The unit was added to the manuscript.

  1. The experimental design is good with ten varieties strawberry and two treatments, however, the measured quality characteristic indicators are too few, there are a total of four indicators. The author should add some other relevant indicators. As for the mentioned Pocket Acid Meter (PAM F5, Atago, Japan) instrument that can quickly measure TA and acid, it is not the innovation of this study. As for the correlation between the study of titratable acidity and acid, it is not significant on strawberries and no convincing results have been obtained.

A: There were five parameters measured (Total anthocyanin content, TSS, TA, pH, acidity or PAM data), with two environments, ten cultivars of strawberries, and 5 harvest dates. There were also four replicates.  This created a large number of samples to process and a huge amount of data. Collecting more data would made the experiment unmanageable. 

  1. If the manuscript includes some pictures with different cultivars in two cultivation conditions will do better.

A: Due to pandemic times, our photo library is limited but we have included two cultivars- Camino Real and Ruby June growing under different cultivation conditions.

Reviewer 2 Report

This manuscript evaluated two environments (open field and high tunnel) in the city of Virginia Beach, and evaluated the fruit quality characteristics (total soluble solids, titratable acidity TA, and total anthocyanin content) of newer strawberry cultivars grown in the annual hill plasticulture system in coastal Virginia climatic conditions. The subject certainly falls within the general scope of Horticulturae journal. However, I have some concerns about the work presented in the MS.

 1. It is suggest to rewrite the Title to contain more comprehensive information and make it more attractive.

 2. The Abstract is not a good writing and needs to be substantially improved. In this study, the results concluded the data of three aspects studies with two environments, ten cultivars of strawberries and 5 harvest seasons. However, a large amount of data are missing. Also, it should clear the significance of the study. In addition, it is better to list some data of the results. Besides, the result of "a new digital meter (Pocket Acidity Meter; PAM) was introduced, which can quickly measure acidity with minimal preparation at the farm." could be listed in the end of The Abstract.

 3. The Introduction was too verbose, and lacked a specific rational and novelty. The authors should provide rational for why study was conducted and in what aspects it is novel from previously conducted similar studies. The logical relation of the research gap and this study was not well listed in the last paragraph of Introduction. It is OK to study several cultivars of strawberries, but the reason of study the two environments (open field and high tunnel) was not explained clearly.

 4. In section of "Materials and Methods", it should give some more detailed measurement methods for the parameters such as the total soluble solids, titratable acidity, the total anthocyanin content. In addition, the soil physical and chemical properties and the growth management must be listed.

 5. The Conclusion paragraph should be more concise. In the current form, it reads more like Abstract.

 6. It is suggested to replace the too old references, such as reference 12, 21 and 25. In addition, it is better to cite literature from journal of Horticulturae.

  In summary, the MS has its merits, it could be considered for publication with Revision.

Author Response

This manuscript evaluated two environments (open field and high tunnel) in the city of Virginia Beach, and evaluated the fruit quality characteristics (total soluble solids, titratable acidity TA, and total anthocyanin content) of newer strawberry cultivars grown in the annual hill plasticulture system in coastal Virginia climatic conditions. The subject certainly falls within the general scope of Horticulturae journal. However, I have some concerns about the work presented in the MS.

  1. It is suggest to rewrite the Title to contain more comprehensive information and make it more attractive.

A: The title has been changed to address the comment

  1. The Abstract is not a good writing and needs to be substantially improved. In this study, the results concluded the data of three aspects studies with two environments, ten cultivars of strawberries and 5 harvest seasons. However, a large amount of data are missing. Also, it should clear the significance of the study. In addition, it is better to list some data of the results. Besides, the result of "a new digital meter (Pocket Acidity Meter; PAM) was introduced, which can quickly measure acidity with minimal preparation at the farm." could be listed in the end of The Abstract.

A: There were five parameters measured (Total anthocyanin content, TSS, TA, pH, acidity or PAM data), with two environments, ten cultivars of strawberries, and 5 harvest dates. There were also four replicates.  This created a large number of samples to process and a huge amount of data.

All data were summarized and presented in the manuscript, except pH. The pH data are not a good representative for taste, therefore, only titratable acidity was presented.

  1. The Introduction was too verbose, and lacked a specific rational and novelty. The authors should provide rational for why study was conducted and in what aspects it is novel from previously conducted similar studies. The logical relation of the research gap and this study was not well listed in the last paragraph of Introduction. It is OK to study several cultivars of strawberries, but the reason of study the two environments (open field and high tunnel) was not explained clearly.

A: New information was added to the introductions that provide rational for why the study was conducted, why in two environments, and why it is novel. Extra text were removed to make it concise.

  1. In section of "Materials and Methods", it should give some more detailed measurement methods for the parameters such as the total soluble solids, titratable acidity, the total anthocyanin content. In addition, the soil physical and chemical properties and the growth management must be listed.

A: The soil physical and chemical properties and the growth management were added to the Materials and Methods. The detailed measurement methods for the total soluble solids, titratable acidity were explained from line 162-173 and for the total anthocyanin content from line 174-185.

  1. The Conclusion paragraph should be more concise. In the current form, it reads more like Abstract.

A: We have revised the wording to make the conclusion more precise.

  1. It is suggested to replace the too old references, such as reference 12, 21 and 25. In addition, it is better to cite literature from journal of Horticulturae.

A: References were updated with three new references and some from MDPI journals.

In summary, the MS has its merits, it could be considered for publication with Revision.

Reviewer 3 Report

Although the manuscript is dealing with some very useful strawberry quality  parameters, it is poorly and unscientifically written.

I will give here some comments, albeit my recommendation is to reject this manuscript. 

1. It seems that manuscript is dealing with two parallel, almost unrelated topics: (1) quality of various cultivars of strawberry fruits grown in two different environment; (2) comparison between titratable acidity (TA) and pocket acidity meter (PAM).

2. Throughout the manuscript, the authors repeated few times the theoretical advantages of TAM over TA.

3. Page 2, line 43: Hectares instead of acres (or both units). 

4. Page 2, line 48: kgs instead of pounds (or both units).

5. Reference 8 is not related to the statement on page 2, line 52.

6. Reference 9 is not related to the statement on page 2, line 64. 

7.  Some pomological information on strawberry cultivars, used in the this research, should be included. Also some info on agro-technical measures etc. should be added.

8. Page 4, line 138: The authors wrote "The total anthocyanin content averaged for..." It is unclear how the values in Table 1 were calculated. 

9. Table 2. Only the anthocyanin contents were presented in this way, whilst TSS and TA were not. Should be uniformed. 

10. Order of the cultivars should be uniform in all Tables.

11. Page 6, line 194-204. Listing the references in this manner makes this paragraph as military report.

12. Page 8 (line233) - page 10 (line 259): I do not see the reason why were the authors dealing with comparison between titratable acidity (TA) and pocket acidity meter (PAM). Seems unrelated to the general idea of the manuscript.

13. Page 11, line 260-264. Already above-written.  

Author Response

Although the manuscript is dealing with some very useful strawberry quality parameters, it is poorly and unscientifically written.

I will give here some comments, albeit my recommendation is to reject this manuscript.

  1. It seems that manuscript is dealing with two parallel, almost unrelated topics: (1) quality of various cultivars of strawberry fruits grown in two different environment; (2) comparison between titratable acidity (TA) and pocket acidity meter (PAM).

A: Thank you for the comment. The two topics are very related as acidity is a main fruit quality parameter. We are offering a new method/tool to measuring acidity to increase the efficiency of harvest for growers. We are suggesting to use (pocket acidity meter, PAM) instead of the titratable acidity which is a tedious, and expensive method.

  1. Throughout the manuscript, the authors repeated few times the theoretical advantages of TAM over TA.

A: We added new information as what does the PAM data is and how it is measured. Meanwhile, the repeated text has been deleted.

  1. Page 2, line 43: Hectares instead of acres (or both units).

A: The correction was made and SI units were used throughout the paper.

  1. Page 2, line 48: kgs instead of pounds (or both units).

A: The correction was made and SI units were used throughout the paper.

  1. Reference 8 is not related to the statement on page 2, line 52.

A: The references were changed and updated.

  1. Reference 9 is not related to the statement on page 2, line 64.

A: The reference has the statement on page 1, introduction, second paragraph.

The references were changed and updated. Please check the reference here: https://www.researchgate.net/publication/267031711_Fruit_quality_of_new_early_ripening_strawberry_cultivars_in_Croatia

  1. Some pomological information on strawberry cultivars, used in the this research, should be included. Also some info on agro-technical measures etc. should be added.

A: Pomological information and agro-technical measures were added to the Materials and Methods.

  1. Page 4, line 138: The authors wrote "The total anthocyanin content averaged for..." It is unclear how the values in Table 1 were calculated.

A: The values were calculated according to Formula (1) in the Methods. Two sentences were added at the end of Experiment Design section to explain the averages.

  1. Table 2. Only the anthocyanin contents were presented in this way, whilst TSS and TA were not. Should be uniformed.

A: Anthocyanin content was presented separately in table 2 because it was affected by the interaction of the genotype and the environment. However, TSS and TA were not affected by the genotype and environment interactions.

  1. Order of the cultivars should be uniform in all Tables.

A: Originally cultivars were ranked for their quality parameter measured. To address the comment, order of the cultivars has changed to be uniform in all table.

  1. Page 6, line 194-204. Listing the references in this manner makes this paragraph as military report.

A: The references were rewritten.

  1. Page 8 (line233) - page 10 (line 259): I do not see the reason why were the authors dealing with comparison between titratable acidity (TA) and pocket acidity meter (PAM). Seems unrelated to the general idea of the manuscript.

A: Three paragraphs were added to the introduction to justify why titratable acidity (TA) and pocket acidity meter (PAM) were measured and why they have been compared.

  1. Page 11, line 260-264. Already above-written.

A: The paragraph was deleted and the information were re-worded in the next paragraph.

Round 2

Reviewer 1 Report

The author has done some modifications or give the explanations in the revised version, I think in can be accepted and published in the Agriculture. 

Reviewer 2 Report

The authors have adequately addressed my concerns and its now ready for publication. 

Reviewer 3 Report

After serious rearrangement of the manuscript, I am glad to confirm that the quality of the paper is significantly improved.